# META-LEARNING UNIVERSAL PRIORS
# USING NON-INJECTIVE NORMALIZING FLOWS

## ABSTRACT

Meta-learning empowers data-hungry deep neural networks to rapidly learn from merely a few samples, which is especially appealing to tasks with small datasets. Critical in this context is the *prior knowledge* accumulated from related tasks. Existing meta-learning approaches typically rely on preselected priors, such as a Gaussian probability density function (pdf). The limited expressiveness of such priors however, hinders the enhanced performance of the trained model when dealing with tasks having exceedingly scarce data. Targeting improved expressiveness, this contribution introduces a *data-driven* prior that optimally fits the provided tasks using a novel non-injective normalizing flow (NNF). Unlike preselected prior pdfs with fixed shapes, the advocated NNF model can effectively approximate a considerably wide range of pdfs. Moreover, compared to conventional injective normalizing flows, the introduced NNF exhibits augmented expressiveness for pdf modeling, especially in high-dimensional spaces. Theoretical analysis underscores the appealing universal approximation capacity of the NNF model. Numerical experiments conducted on three few-shot learning datasets validate the superiority of data-driven priors over the prespecified ones, showcasing its pronounced effectiveness when dealing with extremely limited data resources.

## 1 INTRODUCTION

Advances in deep learning (DL) have boosted the notion of "learning from data" with field-changing performance improvements reported across a wide range of applications (Krizhevsky et al., 2012; Goodfellow et al., 2016; Vaswani et al., 2017). Large-scale DL models with high fitting capacity have documented ability to cope with the "curse of dimensionality" by providing compact low-dimensional representations of high-dimensional data. Nonetheless, these high-capacity models typically require protracted training using massive data records. Humans on the contrary, can perform exceptionally well on tasks such as object recognition or concept comprehension with merely a few samples. How to acquire the learning ability of humans in the DL training processes is thus appealing and imperative for a number of application domains, especially when data are scarce or costly to annotate. Examples of such applications include machine translation (Vaswani et al., 2017), medical imaging (Litjens et al., 2017), and robot manipulations (Levine et al., 2018).

Meta-learning, also referred to as "learning to learn," seeks to gather the *prior knowledge* shared across a set of inter-related tasks, to enable quickly solving an unseen yet related learning task using minimal training samples (Finn et al., 2017). This form of higher-level learning effectively extracts domain-generic inductive biases from prior tasks, which can be subsequently transferred to learn a new task even with limited data. This mirrors the capability that humans excel at — leveraging past experiences to rapidly acquire new skills. Meta-learning holds the promise of yielding powerful priors with which DL models can generalize better, require fewer data for training, and adapt more effectively to new tasks in dynamically changing environments.

Conventional approaches to meta-learning have relied on hand-crafted techniques to extract prior knowledge (Schmidhuber, 1993; 1999). With the advent of DL and growing volume of data, there has been a paradigm shift from such cumbersome procedures towards more efficient data-driven strategies. In particular, the prior information is encoded in hyperparameters, which are shared across tasks and can be fine-tuned using the validation data of all tasks. Utilizing these informative hyperparameters, task-specific learning can be performed even with limited data. Early attempts

adopted a neural network (NN) with its weights serving as the shared hyperparameters (Vinyals et al., 2016; Santoro et al., 2016; Munkhdalai & Yu, 2017). The *task-invariant* NN leverages the shared hyperparameters, and training data per task, to output the *task-specific* model. However, the selection of an appropriate NN architecture is tailored to the choice of the task-specific models. In addition, NNs inherently lack interpretability and robustness due to their "black box" nature.

Unlike NN-based meta-learning, model-agnostic meta-learning (MAML) does not rely on any presumptions about task-specific models (Finn et al., 2017). Instead, it relies on an iterative optimizer to learn the task-specific model. The task-invariant prior information is embodied in the initialization of the optimizer, which is shared across tasks. By learning an informative initialization, task-specific learning can rapidly converge to a local minimum within a few iterations. Interestingly, the initialization generated by MAML can be viewed as a learnable mean of an implicit Gaussian prior probability density function (pdf) over the task-specific model parameters (Grant et al., 2018). Building on MAML, several optimization-based meta-learning algorithms have been advocated to learn different prior pdfs (Li et al., 2017; Park & Oliva, 2019; Lee et al., 2019; Baik et al., 2021; Wang et al., 2023). In addition, theoretical studies have been carried out to further offer insights into these approaches (Franceschi et al., 2018; Rajeswaran et al., 2019; Fallah et al., 2020; Farid & Majumdar, 2021; Zhang et al., 2023). Nevertheless, the prior models of most existing meta-learning methods are confined to preselected pdfs, such as the Gaussian one, and thus have limited expressiveness, meaning fitting ability. Consequently, generalizing meta-learning to domains that deal with scarce datasets, and need sophisticated priors, remains a challenging and largely uncharted territory.

To improve the prior expressiveness in meta-learning, this contribution puts forth what we term non-injective normalizing flow (NNF) model, which enables learning a universal data-driven prior from related tasks. The contribution of the resultant method named MetaNNF is threefold:

i) By waiving the injectivity constraint of normalizing flows (NFs) (Rezende & Mohamed, 2015), our novel NNF model is proven capable of mapping a known source pdf to an arbitrary target pdf. This markedly enhances expressiveness of NFs, especially in high-dimensional spaces.

ii) Theoretical analysis is provided to demonstrate that the proposed parametric NNF can approximate a broad spectrum of pdfs, that in turn enables versatile plug-in prior pdfs for meta-learning. Moreover, this parametric NNF inherently provides a task-invariant initialization, rather nicely eliminating the need for its explicit learning.

iii) Numerical tests on three benchmark few-shot learning datasets corroborate our theoretical analysis, and underscore the superior prior expressiveness of the proposed MetaNNF method compared to meta-learning approaches with prespecified pdfs.

## 2 PROBLEM SETUP

Meta-learning relies on task-invariant prior information from a collection of $T$ given tasks (indexed by $t = 1, \ldots, T$), to deal with data-limited settings. For each $t$, there is an annotated dataset $\mathcal{D}_t := \{(\mathbf{x}_t^n, y_t^n)\}_{n=1}^{N_t}$ consisting of $N_t$ (data, label) pairs. The dataset is divided into a training subset $\mathcal{D}_t^{\mathrm{trn}} \subset \mathcal{D}_t$, and a validation subset $\mathcal{D}_t^{\mathrm{val}} := \mathcal{D}_t \setminus \mathcal{D}_t^{\mathrm{trn}}$. In addition, a new task indexed by $\star$ is also provided, with its training set $\mathcal{D}_\star^{\mathrm{trn}}$, and an unannotated test set $\mathcal{D}_\star^{\mathrm{tst}} := \{\mathbf{x}_\star^n\}_{n=1}^{N_\star^{\mathrm{tst}}}$ for which the corresponding labels $\{y_\star^n\}_{n=1}^{N_\star^{\mathrm{tst}}}$ are to be inferred. The major premise of meta-learning is that the aforementioned tasks are related through their underlying data distributions or problem structures. This relationship makes it feasible to employ a unified large-scale model such as a deep NN to fit all tasks, with each task tailored by its specific model parameter $\phi_t \in \mathbb{R}^d$. However, as the cardinality $|\mathcal{D}_t^{\mathrm{trn}}|$ can be much smaller than $d$, directly optimizing $\phi_t$ over $\mathcal{D}_t^{\mathrm{trn}}$ could readily lead to overfitting.

Meta-learning addresses this issue by capitalizing on the relationships among tasks. Specifically, since $T$ is considerably large in meta-learning, a *task-invariant* prior can be extracted to capture knowledge across tasks, thereby facilitating the data-limited per-task training. This nested structure of prior extraction and per-task training lends itself to a *bilevel optimization* problem. The inner-level (task-level) optimizes the per-task parameter $\phi_t$ using $\mathcal{D}_t^{\mathrm{trn}}$, and the prior provided by outer-level, while the outer-level (meta-level) evaluates the trained $\{\phi_t\}_{t=1}^{T}$ using $\{\mathcal{D}_t^{\mathrm{val}}\}_{t=1}^{T}$, and refines the prior parameterized by $\boldsymbol{\theta} \in \mathbb{R}^D$, where it is possible to have $D \gg d$.

The bilevel optimization objective of meta-learning can be expressed as

$$\min_{\boldsymbol{\theta}} \sum_{t=1}^{T} \mathcal{L}(\boldsymbol{\phi}_t^*(\boldsymbol{\theta}); \mathcal{D}_t^{\text{val}}) \tag{1a}$$

$$\text{s.t. } \boldsymbol{\phi}_t^*(\boldsymbol{\theta}) = \operatorname*{argmin}_{\boldsymbol{\phi}_t} \mathcal{L}(\boldsymbol{\phi}_t; \mathcal{D}_t^{\text{trn}}) + \mathcal{R}(\boldsymbol{\phi}_t; \boldsymbol{\theta}), \ t = 1, \dots, T \tag{1b}$$

where the loss function $\mathcal{L}$ assesses the fit of a task-specific model to a designated dataset, and the regularizer $\mathcal{R}$ quantifies the impact of task-invariant prior. From the Bayesian viewpoint, $\mathcal{L}(\boldsymbol{\phi}_t; \mathcal{D}_t^{\text{trn}}) = -\log p(\mathbf{X}_t^{\text{trn}} | \boldsymbol{\phi}_t; \mathbf{y}_t^{\text{trn}})$ can be interpreted as the negative log-likelihood (nll), and $\mathcal{R}(\boldsymbol{\phi}_t; \boldsymbol{\theta}) = -\log p(\boldsymbol{\phi}_t; \boldsymbol{\theta})$ is the negative log-prior (nlp), where $\mathbf{X}_t^{\text{trn}}$ denotes the matrix collecting all the data vectors in $\mathcal{D}_t^{\text{trn}}$, and $\mathbf{y}_t^{\text{trn}}$ is the corresponding label vector. Using Bayes' rule, it follows that $\boldsymbol{\phi}_t^* = \operatorname{argmax}_{\boldsymbol{\phi}} p(\boldsymbol{\phi}_t | \mathbf{y}_t^{\text{trn}}; \mathbf{X}_t^{\text{trn}}, \boldsymbol{\theta})$ is the maximum a posteriori (MAP) estimator.

Unfortunately, the global optimum $\boldsymbol{\phi}_t^*$ in (1b) is generally unreachable when the postulated model is a nonlinear function of $\boldsymbol{\phi}_t$. Hence, a feasible alternative is to rely on an approximate solver $\hat{\boldsymbol{\phi}}_t \approx \boldsymbol{\phi}_t^*$ obtained by a tractable optimizer. Depending on how the alternative solver is acquired, meta-learning algorithms can be categorized as either NN- or optimization-based ones. The former harnesses an NN optimizer $\hat{\boldsymbol{\phi}}_t = \text{NN}(\mathcal{D}_t^{\text{trn}}; \boldsymbol{\theta})$ to model the training process that maps $\mathcal{D}_t^{\text{trn}}$ to $\hat{\boldsymbol{\phi}}_t$, with the sought prior encoded in the NN's learnable weights $\boldsymbol{\theta}$ (Ravi & Larochelle, 2017; Gordon et al., 2019). Despite the effectiveness of NN optimizers in fitting complex mappings, it is hard to decipher the learned prior due to their black-box nature. To improve the interpretability and robustness of the approximate solver, optimization-based meta-learning decodes the "tractable optimizer" as a cascade of a few optimization iterations. The prior is captured by the shared hyperparameters of the optimizer. The first effort towards this direction is termed MAML (Finn et al., 2017), which relies on a $K$-step gradient descent (GD) optimizer

$$\boldsymbol{\phi}_t^{(k)}(\boldsymbol{\theta}) = \boldsymbol{\phi}_t^{(k-1)}(\boldsymbol{\theta}) - \nabla \mathcal{L}(\boldsymbol{\phi}_t^{(k-1)}(\boldsymbol{\theta}); \mathcal{D}_t^{\text{trn}}), \quad k = 1, \dots, K \tag{2}$$

where $K$ denotes a preselected small number of iterations, task-invariant initialization $\boldsymbol{\phi}_t^{(0)} = \boldsymbol{\phi}^{(0)} = \boldsymbol{\theta}$ parameterizes the prior information, and $\hat{\boldsymbol{\phi}}_t = \boldsymbol{\phi}_t^{(K)}$ gives the desired approximate solver. Interestingly, despite the absence of an explicit regularization term (that is, $\mathcal{R}(\boldsymbol{\phi}_t; \boldsymbol{\theta}) = 0$), it has been shown that MAML's GD solver (2) satisfies (Grant et al., 2018)

$$\hat{\boldsymbol{\phi}}_t(\boldsymbol{\theta}) \approx \boldsymbol{\phi}_t^*(\boldsymbol{\theta}) = \operatorname*{argmin}_{\boldsymbol{\phi}_t} \mathcal{L}(\boldsymbol{\phi}_t; \mathcal{D}_t^{\text{trn}}) + \frac{1}{2} \|\boldsymbol{\phi}_t - \boldsymbol{\phi}^{(0)}\|_{\boldsymbol{\Lambda}_t}^2, \ t = 1, \dots, T$$

where the precision matrix $\boldsymbol{\Lambda}_t$ is determined by $\alpha$, $K$, and $\nabla^2 \mathcal{L}(\boldsymbol{\phi}^{(0)}; \mathcal{D}_t^{\text{trn}})$. This observation indicates MAML's optimizer approximately amounts to an implicit Gaussian prior $p(\boldsymbol{\phi}_t; \boldsymbol{\theta}) \approx \mathcal{N}(\boldsymbol{\phi}_t; \boldsymbol{\phi}^{(0)}, \boldsymbol{\Lambda}_t^{-1})$, with the shared initialization $\boldsymbol{\phi}^{(0)} = \boldsymbol{\theta}$ serving as its mean vector.

Building upon MAML, various methods have been investigated to learn different prior pdfs in both implicit and explicit forms. For example, recent advances further render the precision matrix learnable by replacing it with a $\boldsymbol{\Lambda}$ that is common across tasks. Letting $\boldsymbol{\theta_\Lambda}$ denote the parameter of $\boldsymbol{\Lambda}$, the prior parameter is thus augmented as $\boldsymbol{\theta} := [\boldsymbol{\phi}^{(0)\top}, \boldsymbol{\theta_\Lambda}^\top]$, where $\top$ denotes transposition. However, a complete parametrization of $\boldsymbol{\Lambda}$ would result in $\boldsymbol{\theta}$ having prohibitively high dimensionality, that is, $D = \mathcal{O}(d^2)$. To ensure scalability with respect to $D$, $\boldsymbol{\Lambda}$ should have a sufficiently simple structure such as isotropic (Rajeswaran et al., 2019), diagonal (Li et al., 2017), and or block diagonal (Lee & Choi, 2018; Park & Oliva, 2019) matrices. Inspired by transfer learning, one can instead split the model into an embedding "body" and a classifier/regressor "head," and learn their priors independently; that is, with $\boldsymbol{\phi}_t^{\text{body}}$ and $\boldsymbol{\phi}_t^{\text{head}}$ denoting the corresponding partitions of $\boldsymbol{\phi}_t$, the prior is presumed factorable as $p(\boldsymbol{\phi}_t; \boldsymbol{\theta}) = p(\boldsymbol{\phi}_t^{\text{body}}; \boldsymbol{\theta}) p(\boldsymbol{\phi}_t^{\text{head}}; \boldsymbol{\theta})$. On the one hand, the head typically has a nontrivial prior such as the Gaussian one (Bertinetto et al., 2019; Lee et al., 2019). On the other hand, the body's prior is intentionally restricted to a degenerate pdf $p(\boldsymbol{\phi}_t^{\text{body}}; \boldsymbol{\theta}) := \delta(\boldsymbol{\phi}_t^{\text{body}} - \boldsymbol{\phi}^{\text{body}})$, where $\boldsymbol{\phi}^{\text{body}}$ is a subvector of $\boldsymbol{\theta}$, and $\delta(\cdot)$ is the Dirac delta function. This eliminates the need for optimizing $\boldsymbol{\phi}_t^{\text{body}}$ in (1b), thus markedly lowering the overall complexity for solving (1). Although freezing the body in (1b) allows for escalating the dimension of $\boldsymbol{\phi}_t^{\text{body}}$, it often leads to degraded empirical performance (Raghu et al., 2020) compared to the full update (2).

## 3 META-LEARNING USING NON-INJECTIVE NORMALIZING FLOWS

Existing meta-learning algorithms rely on a *preselected* pdf to parameterize the prior. However, the chosen pdf can have limited expressiveness; that is, it may have insufficient ability to offer an accurate fit. Consider for instance a preselected Gaussian prior pdf, which is inherently unimodal, symmetric, log-concave, and infinitely differentiable by definition. Such a prior may not be well-suited for tasks with multimodal or asymmetric parametric pdfs. In this work, we propose to learn a *data-driven* prior pdf that optimally fits the given tasks using a novel non-injective normalizing flow (NNF) model. We thus term the proposed method as Meta-learning with Non-injective Normalizing Flows (MetaNNF). Injective NFs and their applications in pdf estimations will be first elaborated. All the proofs are delegated to the Appendix.

### 3.1 PDF ESTIMATION VIA INJECTIVE NORMALIZING FLOWS

NFs were introduced in (Rezende & Mohamed, 2015) as a surrogate variational model to approximately infer intractable posterior pdfs. Recently, they have been shown also effective in estimating prior pdfs from a set of unannotated samples (Dinh et al., 2015; Germain et al., 2015; Dinh et al., 2017). The formulation of NFs relies on the well-known change-of-variable formula. Given a continuous random vector $\mathbf{Z} \in \mathbb{R}^d$ with known prior pdf $p_{\mathbf{Z}} : \mathbb{R}^d \mapsto \mathbb{R}^+ \cup \{0\}$, and a bijection $f : \mathbb{R}^d \mapsto \mathbb{R}^d$, the transformed $\mathbf{Z}' := f(\mathbf{Z})$ is also a continuous random vector with analytical pdf

$$p_{\mathbf{Z}'}(\mathbf{z}') = p_{\mathbf{Z}}(f^{-1}(\mathbf{z}')) \left| \det J_{f^{-1}}(\mathbf{z}') \right| = \frac{p_{\mathbf{Z}}(f^{-1}(\mathbf{z}'))}{\left| \det J_f(\mathbf{z}') \right|} \text{ (a.e.)} \tag{3}$$

where $J_f(\mathbf{z}')$ denotes the Jacobian of $f$ at $\mathbf{z}' \in \mathbb{R}^d$, $\det$ is the determinant, and $\det J_f \neq \mathbf{0}$ almost everywhere (a.e.) for bijective $f$. To ensure the invertibility of $f$, a prudent choice is to model it as a composition of a sequence of bijective functions $f = f_1 \circ f_2 \circ \ldots \circ f_n$. By optimizing the (parametric) bijection $f$, (3) can be adjusted to approximate a target pdf $q$. In Bayesian inference (Rezende & Mohamed, 2015), $q$ is an intractable posterior, and $f$ is optimized to minimize the KL-divergence between $p_{\mathbf{Z}'}$ and $q$, or equivalently, maximize the so-termed evidence lower bound (ELBO). For density estimation (Dinh et al., 2015), the wanted $q$ is an unknown prior pdf, while $f$ is acquired via maximum likelihood training. The obtained $f$ can be leveraged in two important applications: i) probability estimation $p_{\mathbf{Z}'}(\mathbf{v}) \approx q(\mathbf{v})$ for a given sample $\mathbf{v} \sim q$ using (3), and ii) generation of a sample $\mathbf{z}' = f(\mathbf{z}), \mathbf{z} \sim p_{\mathbf{Z}}$ for which $p_{\mathbf{Z}'} \approx q$.

When $d = 1$, the probability integral transform (PIT) suggests that, the optimal $f^* = Q^{-1} \circ P_{\mathbf{Z}}$ leads to precisely $P_{\mathbf{Z}'} = Q$ a.e., where $Q$, $P_{\mathbf{Z}}$ and $P_{\mathbf{Z}'}$ are the cumulative distribution functions (cdfs) corresponding to $q$, $p_{\mathbf{Z}}$ and $p_{\mathbf{Z}'}$, and $q > 0$ a.e. ensures $Q$ is bijective. The resultant cdf $P_{\mathbf{Z}'} = P_{\mathbf{Z}} \circ f^{*-1}$ is a pushforward measure, also notated as $P_{\mathbf{Z}'} = f^* \# P_{\mathbf{Z}}$. In high-dimensional spaces ($d > 1$) however, the existence of such an $f^*$ may not hold due to the invertibility assumption of $f^*$, even when $q > 0$ a.e.; see examples in e.g., (Kong & Chaudhuri, 2020, Section 4).

### 3.2 IMPROVED PDF ESTIMATION VIA NON-INJECTIVE NORMALIZING FLOWS

To improve the fitting capacity of NFs for generic $q$, especially those in high-dimensional spaces, the fresh idea of this work is to forgo the injectivity assumption on $f$. In doing so, we can generalize the PIT to an arbitrary $q$ even in a high-dimension space, as illustrated in the following theorem.

**Theorem 3.1** (Multivariate PIT). *Consider measurable space $(\mathbb{R}^d, \mathcal{B}(\mathbb{R}^d))$ where $\mathcal{B}$ is the Borel $\sigma$-algebra. Let $P_{\mathbf{Z}} : \mathbb{R}^d \mapsto [0, 1]$ be the cdf of continuous random vector $\mathbf{Z} := [Z_1, \ldots, Z_d]^\top$ with $\{Z_i\}_{i=1}^d$ mutually independent. For any differentiable a.e. cdf $Q : \mathbb{R}^d \mapsto [0, 1]$, there exists a weakly increasing function $f^* : \mathbb{R}^d \mapsto \mathbb{R}^d$ for which the random vector $\mathbf{Z}' := f^*(\mathbf{Z})$ has cdf*

$$P_{\mathbf{Z}'} = Q \text{ (a.e.)}. \tag{4}$$

**Remark 3.2** (Choice of source distribution). In the theorem, the prior distribution for the source random vector $\mathbf{Z}$ can be chosen arbitrarily, as if it has mutually independent entries. Popular choices include standard Gaussian $\mathcal{N}(\mathbf{0}_d, \mathbf{I}_d)$ and uniform $\text{Uniform}([0, 1]^d)$.

**Remark 3.3** (Challenges in the proof). In Theorem 3.1, $Q$ is a projection from $\mathbb{R}^d$ to $[0, 1] \subset \mathbb{R}$. Compared to the univariate PIT, this reduction of projected dimensionality generally renders $Q^{-1}$ nonunique. On the other hand, applying the univariate PIT on each dimension of $\mathbf{Z}$ will lead to a

$\mathbf{Z}'$ with mutually independent entries. To capture the mutual dependencies in $Q$, our proof utilizes a series of conditional cdfs derived from $Q$ to recursively construct each dimension of $f^*$, which enables us to match the copulas of $P_{\mathbf{Z}}$ and $Q$; see Appendix A for more details.

**Remark 3.4** (Comparison with injective NFs). While conventional NFs (3) require $J_f \neq \mathbf{0}$ a.e. (typically $J_f \succ \mathbf{0}$ (Germain et al., 2015; Dinh et al., 2017)) to ensure the injectivity of $f$, Theorem 3.1 relaxes this assumption to $J_f \succeq 0$. This allows $f$ to be non-injective and thus enables $\mathbf{Z}' = f(\mathbf{Z})$ to match an arbitrary target distribution (even discrete one) in a high-dimensional space. It is worth mentioning that the mild assumption on the differentiability of $Q$ is merely used to guarantee the existence of $q$, which can be easily satisfied by most cdfs of interest. However, one limitation of the advocated NNF is that it generally has no analytical solution for the resultant surrogate pdf

$$p_{\mathbf{Z}'}(\mathbf{z}') = \int_{\mathbb{R}^d} p_{\mathbf{Z}}(\mathbf{z})\delta[\mathbf{z}' - f(\mathbf{z})]d\mathbf{z}. \tag{5}$$

As a remedy, efficient numerical integration can be performed to estimate $p_{\mathbf{Z}'}$ when $d$ is small. Consequently, the proposed NNF is particularly effective for sample generation rather than density estimation, which is similar to (Kingma et al., 2016).

While Theorem 3.1 suggests the existence of the optimal $f^*$ that incurs the exact match $p_{\mathbf{Z}'} = q$, the expression for such an $f^*$ relies on the sought $q$, which is typically intractable or unknown. Therefore, a feasible alternative is to resort to a tractable parametric $f(\cdot; \boldsymbol{\theta}_f)$, which approximates $f^*$ by learning $\boldsymbol{\theta}_f$ from the provided data. To streamline the discussion, we will focus exclusively on Sylvester NF (van den Berg et al., 2018) in the following sections, but our analysis can be readily generalized to other NFs; see Remark 3.9. Sylvester NF were introduced in (van den Berg et al., 2018) to improve the expressiveness of planar NF (Rezende & Mohamed, 2015) by increasing its "width". In particular, Sylvester NF adopts the form

$$f(\mathbf{Z}; \boldsymbol{\theta}_f) := \mathbf{Z} + \mathbf{A}\sigma(\mathbf{B}\mathbf{Z} + \mathbf{c}), \ \mathbf{Z} \in \mathbb{R}^d \tag{6}$$

where $\mathbf{A} \in \mathbb{R}^{d \times m}, \mathbf{B} \in \mathbb{R}^{m \times d}, \mathbf{c} \in \mathbb{R}^m$ are learnable weights with $m$ being the number of hidden neurons (a.k.a. width), $\sigma$ is an entry-wise nonlinear operator, and $\boldsymbol{\theta}_f := [\text{vec}(\mathbf{A})^\top, \text{vec}(\mathbf{B})^\top, \mathbf{c}^\top]^\top$. It can be easily verified that the Sylvester NF boils down to the planar one when $m = 1$. Akin to other NFs, one can also increase the "depth" of the flows by stacking multiple Sylvester NF layers into a chain $f_1 \circ f_2 \circ \ldots \circ f_n$. The next theorem states that, the optimal $f^*$ can be approximated to arbitrary precision using a sufficiently wide one-layer Sylvester NF.

**Definition 3.5.** A random vector on $\mathbb{R}^d$ is said to be tail-convergent if i) it has a pdf $p : \mathbb{R}^d \mapsto \mathbb{R}^+ \cup \{0\}$, and ii) for $\forall \epsilon > 0$ there exists a bounded $E \subset \mathbb{R}^d$ for which

$$\int_{\mathbb{R}^d \backslash E} p < \epsilon. \tag{7}$$

**Theorem 3.6** (Universal approximation via non-injective Sylvester NFs). *Let $P_{\mathbf{Z}}$ denote the cdf of tail-convergent continuous random vector $\mathbf{Z} \in \mathbb{R}^d$ with mutually independent entries, and $Q$ a Lipschitz cdf of a tail-convergent random vector. For any $\epsilon > 0$, there exists cdfs $\tilde{P}, \tilde{Q}$ for which the corresponding pdfs $\tilde{p}, \tilde{q}$ vanishes outside compact sets $E_p, E_q$, and*

$$|P_{\mathbf{Z}}(\mathbf{v}) - \tilde{P}(\mathbf{v})| < \epsilon, \ |Q_{\mathbf{Z}}(\mathbf{v}) - \tilde{Q}(\mathbf{v})| < \epsilon, \ \forall \mathbf{v} \in \mathbb{R}^d. \tag{8}$$

*Moreover, let $E \subseteq E_p$ be any set on which the optimal $f^*$ matching $\tilde{P}_{\mathbf{Z}}$ to $\tilde{Q}$ (cf. Theorem 3.1) is injective. There exists a non-injective Sylvester NF $f$ and a zero-measure set $E_0$, such that*

$$|f(\mathbf{Z}) - f^*(\mathbf{Z})| < \epsilon, \ \forall \mathbf{Z} \in E_p \backslash E_0, \tag{9a}$$

$$|P_{\mathbf{Z}}(\mathbf{z}) - Q \circ f(\mathbf{z})| < \epsilon, \ \forall \mathbf{z} \in E \backslash E_0. \tag{9b}$$

**Remark 3.7** (Approximation of pushforward). We have shown that when $f^*$ is injective, the cdf of the optimally transformed $\mathbf{Z}' = f^*(\mathbf{Z})$ can be written as a pushforward $Q = P_{\mathbf{Z}'} = P_{\mathbf{Z}} \circ f^{*-1}$. Likewise, this relationship remains valid when restricting $f^*$ to a set $E$ on which $f^*$ is injective. However, since the Sylvester NF $f$ may not be injective on $E$, one cannot directly compare $Q$ with $P_{\mathbf{Z}} \circ f^{-1}$. Fortunately, this pushforward can be equivalently written as $P_{\mathbf{Z}}(\mathbf{z}) = Q \circ f^*(\mathbf{z}), \ \forall \mathbf{z} \in E$; see Lemma B.1 in the Appendix. Utilizing this alternative relationship, Theorem 3.6 states that the Sylvester NF $f$ not only approximates $f^*$ a.e. on $E_p$, but also results in pushforward approximation $P_{\mathbf{Z}} \approx Q \circ f$ a.e. on $E$.

---

**Algorithm 1:** MetaNNF algorithm

---

**Input:** $\{\mathcal{D}_t\}_{t=1}^T$, step sizes $\alpha$ and $\beta$, batch size $B$, and maximum iterations $K$ and $R$.

**Initialization:** randomly initialize $\boldsymbol{\theta}_f^{(0)}$.

1  **for** $r = 1, \ldots, R$ **do**
2  $\quad$ Randomly sample a mini-batch $\mathcal{T}^{(r)} \subset \{1, \ldots, T\}$ of cardinality $B$;
3  $\quad$ **for** $t \in \mathcal{T}^{(r)}$ **do**
4  $\quad\quad$ Initialize $\mathbf{z}_t^{(0)} = \arg\min_{\mathbf{z}_t} \mathcal{R}_{\mathbf{Z}}(\mathbf{z}_t)$;
5  $\quad\quad$ **for** $k = 1, \ldots, K$ **do**
6  $\quad\quad\quad$ Descend
$\quad\quad\quad \mathbf{z}_t^{(k)}(\boldsymbol{\theta}_f^{(r-1)}) = \mathbf{z}_t^{(k-1)} - \alpha \nabla_{\mathbf{z}_t^{(k-1)}} [\mathcal{L}(f(\mathbf{z}_t^{(k-1)}; \boldsymbol{\theta}_f^{(r-1)}); \mathcal{D}_t^{\text{trn}}) + \mathcal{R}_{\mathbf{Z}}(\mathbf{z}_t^{(k-1)})]$;
7  $\quad\quad$ **end**
8  $\quad\quad$ Approximate solver $\mathbf{z}_t(\boldsymbol{\theta}_f^{(r-1)}) = \mathbf{z}_t^{(K)}(\boldsymbol{\theta}_f^{(r-1)})$;
9  $\quad$ **end**
10 $\quad$ Update $\boldsymbol{\theta}_f^{(r)} = \boldsymbol{\theta}_f^{(r-1)} - \beta \frac{1}{|\mathcal{T}^{(r)}|} \sum_{t \in \mathcal{T}^{(r)}} \nabla_{\boldsymbol{\theta}_f^{(r-1)}} \mathcal{L}(f(\mathbf{z}_t(\boldsymbol{\theta}_f^{(r-1)}); \boldsymbol{\theta}_f^{(r-1)}); \mathcal{D}_t^{\text{val}})$;
11 **end**

**Output:** $\hat{\boldsymbol{\theta}}_f = \boldsymbol{\theta}_f^{(R)}$.

---

**Remark 3.8** (Mild assumptions). The assumptions in Theorem 3.6 are mild and common. In particular, tail-convergence only requires the probability of large deviation diminishing to $0$ as the norm of the random vector goes to $+\infty$, while imposing no specific constraint on the decaying rate. This assumption can be easily satisfied by a wide family of distributions, even including the heavily-tailed ones. Under this benign assumption, (8) suggests $P_{\mathbf{Z}}$ and $Q$ can be approximated by alternatives $\tilde{P}, \tilde{Q}$ with pdfs $\tilde{p}, \tilde{q}$ having truncated tails. This is crucial to universal approximation, which typically requires $f^*$ to be bounded or Lebesgue integrable (Cybenko, 1989). Moreover, the Lipschitzness of $Q$ is solely utilized to ensure the boundness of its gradient, namely the pdf $q$. This can be also readily met by most practical cdfs.

**Remark 3.9** (Generalization to other NFs). Although Theorem 3.6 primarily focuses on one-layer Sylvester NFs, similar analysis for other NFs can be acquired by employing different universal approximation models. For instance, results for multi-layer planar NFs and multi-layer Sylvester NFs can be respectively established leveraging (Lin & Jegelka, 2018) and (Lu et al., 2017).

**Remark 3.10** (Influence of $\epsilon$). It is worth noting that the width $m$ of the Sylvester NF depends on $\epsilon$ as well as the optimal $f^*$. Smaller $\epsilon$ typically leads to larger $m$. Additionally, the nonlinearity $\sigma$ must be sigmoidal; see Definition B.3 in the Appendix for details.

## 3.3 META-LEARNING UNIVERSAL PRIORS VIA META-NNF

Next, we elucidate how universal priors can be learned in meta-learning by harnessing the proposed NNF model. Different from existing works that rely on prespecified prior forms such as Gaussian pdfs, the novel concept of this work is to learn a data-driven prior that optimally conforms with the given tasks. This is achieved by transforming the random vector $\mathbf{Z} \in \mathbb{R}^d$ with a known prior $p_{\mathbf{Z}}$ to $\mathbf{Z}' = f(\mathbf{Z}; \boldsymbol{\theta}_f)$, whose pdf is given by (5). This $p_{\mathbf{Z}'}$ acts as a surrogate model for the unknown $p(\boldsymbol{\phi}_t; \boldsymbol{\theta})$, and learning the prior parameter $\boldsymbol{\theta}$ thus boils down to optimization of the transformation parameter $\boldsymbol{\theta}_f$. Nevertheless, as discussed in Remark 3.4, $p_{\mathbf{Z}'}$ typically has no close-form expression when $f$ is non-injective. Therefore, instead of directly optimizing $\boldsymbol{\phi}_t$, we propose to optimize the latent vector $\mathbf{z}_t$ corresponding to $\boldsymbol{\phi}_t = f(\mathbf{z}_t)$, which yields

$$\min_{\boldsymbol{\theta}_f} \sum_{t=1}^T \mathcal{L}(f(\mathbf{z}_t^*(\boldsymbol{\theta}_f); \boldsymbol{\theta}_f); \mathcal{D}_t^{\text{val}}) \tag{10a}$$

$$\text{s.t. } \mathbf{z}_t^*(\boldsymbol{\theta}_f) = \arg\min_{\mathbf{z}_t} \mathcal{L}(f(\mathbf{z}_t; \boldsymbol{\theta}_f); \mathcal{D}_t^{\text{trn}}) + \mathcal{R}_{\mathbf{Z}}(\mathbf{z}_t), \, t = 1, \ldots, T \tag{10b}$$

where $\mathcal{R}_{\mathbf{Z}}(\mathbf{z}_t) := -\log p_{\mathbf{Z}}(\mathbf{z}_t)$ is the nlp regularizer, and $\mathbf{z}_t^*$ is thus the MAP estimator for $\mathbf{z}_t$.

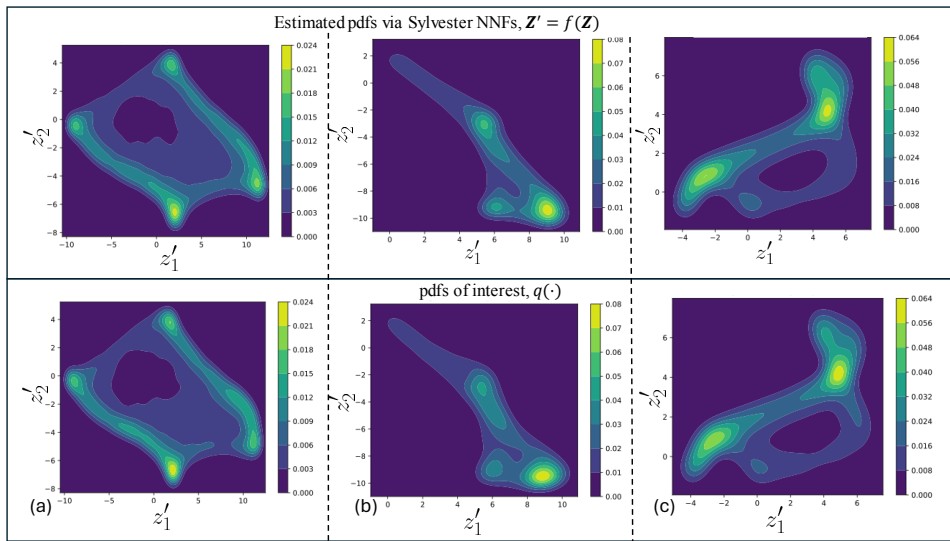

Figure 1: Transforming a standard Gaussian pdf into multi-modal target pdfs using Sylvester NNFs.

Similar to (2), the global task-level minimizer $\mathbf{z}_t^*$ is generally infeasible to attain. Hence, a tractable alternative is to rely on an approximate GD solver. Interestingly, our formulation (10) naturally offers a convenient initialization using the *maximum a priori estimator*

$$\mathbf{z}_t^{(0)} = \operatorname*{argmax}_{\mathbf{z}_t} p_{\mathbf{Z}}(\mathbf{z}_t) = \operatorname*{argmin}_{\mathbf{z}_t} \mathcal{R}_{\mathbf{Z}}(\mathbf{z}_t), \ t = 1, \dots, T \tag{11}$$

As an example, choosing $p_{\mathbf{Z}} = \mathcal{N}(\mathbf{0}_d, \mathbf{I}_d)$ automatically gives $\mathbf{z}_t^{(0)} = \mathbf{0}_d$ and the corresponding $\boldsymbol{\phi}_t^{(0)} = f(\mathbf{0}_d; \boldsymbol{\theta}_f)$. This elegantly removes the need for separately learning the task-invariant initialization $\boldsymbol{\phi}^{(0)}$, which is exactly the maximum a priori estimator of the preselected Gaussian prior pdf $p(\boldsymbol{\theta}_t; \boldsymbol{\theta}) = \mathcal{N}(\boldsymbol{\phi}^{(0)}, \boldsymbol{\Lambda}_t)$. In fact, the task-invariant initialization reflects our optimal guess of $\boldsymbol{\phi}_t$ before accessing any task-specific data, and can be naturally derived by maximizing the prior pdf.

To this end, (10) can be solved using a standard alternating optimizer. The resultant MetaNNF algorithm is listed step-by-step in Algorithm 1, where the inner-level (10b) and outer-level (10a) are respectively optimized using $K$-step GD and mini-batch stochastic GD.

## 4 NUMERICAL TESTS

Here we test and showcase the empirical superiority of MetaNNF on both synthetic and real datasets. Our codes are run on a server equipped with an Intel Core i7-12700 CPU, and an NVIDIA RTX A5000 GPU. All datasets descriptions and hyperparameter setups are deferred to the Appendix D.

### 4.1 TESTS WITH TOY DATA

Here, we investigate an intricate yet interesting scenario to demonstrate the efficacy of NNFs to approximate complex multi-modal pdfs in two-dimensional (2D) settings. The primary objective is to transform a standard Gaussian random vector $\mathbf{Z} \sim \mathcal{N}(\mathbf{0}_{2 \times 1}, \mathbf{I}_{2 \times 2})$ into multi-modal complex pdfs. The outcomes of this experiment are presented in Fig. 1. The lower row displays the ground-truth pdfs $q$ of interest, while the upper row showcases the numerically estimated pdfs of the transformed random vector $\mathbf{Z}' = f(\mathbf{Z})$, where $f$ is a Sylvester NNF, and the pdf of $\mathbf{Z}'$ is estimated via 5. As clearly evidenced in these results, the advocated NNFs exhibit their capability to effectively convert a basic Gaussian distribution into intricate multi-modal distributions in 2D. The expressiveness of non-injective Sylvester NFs in approximating 1D mixture of Gaussians is postponed to Appendix C.

### 4.2 PERFORMANCE EVALUATION USING REAL DATA

Next, the empirical performance of MetaNNF is assessed on three real datasets for meta-learning.

Table 1: Performance comparison of MetaNNF against meta-learning methods having different priors on miniImageNet. For fairness, only methods with a 4-block CNN backbone have been included. The highest accuracy as well as the mean accuracies within its $95\%$ confidence interval are bolded.

| Method | Prior model | 5-class miniImageNet | |
| --- | --- | --- | --- |
| | | 1-shot (%) | 5-shot (%) |
| Meta-LSTM (Ravi & Larochelle, 2017) | RNN-based | $43.44 \pm 0.77$ | $60.60 \pm 0.71$ |
| MAML (Finn et al., 2017) | implicit Gaussian | $48.70 \pm 1.84$ | $63.11 \pm 0.92$ |
| MetaSGD (Li et al., 2017) | diagonal Gaussian | $50.47 \pm 1.87$ | $64.03 \pm 0.94$ |
| R2D2 (Bertinetto et al., 2019) | degenerate body & Gaussian head | $51.8 \pm 0.2$ | $68.4 \pm 0.2$ |
| MC (Park & Oliva, 2019) | block-diagonal Gaussian | $54.08 \pm 0.93$ | $67.99 \pm 0.73$ |
| Warp-MAML (Flennerhag et al., 2020) | Gaussian | $52.3 \pm 0.8$ | $68.4 \pm 0.6$ |
| MAML + L2F (Baik et al., 2020) | implicit Gaussian | $52.10 \pm 0.50$ | $69.38 \pm 0.46$ |
| MeTAL (Baik et al., 2021) | implicit Gaussian | $52.63 \pm 0.37$ | $70.52 \pm 0.29$ |
| Minimax-MAML (Wang et al., 2023) | inverted Gaussian & entropy | $51.70 \pm 0.42$ | $68.41 \pm 1.28$ |
| MAML + MetaNNF (ours) | NNF-based | $\mathbf{57.74 \pm 1.47}$ | $70.72 \pm 0.70$ |
| MetaSGD + MetaNNF (ours) | | $\mathbf{59.10 \pm 1.52}$ | $\mathbf{71.48 \pm 0.68}$ |

Table 2: Performance comparison using the WRN-28-10 features (Rusu et al., 2019). $^\dagger$ indicates that both training and validation tasks are used in the training phase of meta-learning.

| Method | Crop | 5-class miniImageNet | | 5-class tieredImageNet | |
| --- | --- | --- | --- | --- | --- |
| | | 1-shot (%) | 5-shot (%) | 1-shot (%) | 5-shot (%) |
| MetaSGD (Li et al., 2017) | center | $56.58 \pm 0.21$ | $68.84 \pm 0.19$ | $59.75 \pm 0.25$ | $69.04 \pm 0.22$ |
| LEO$^\dagger$ (Rusu et al., 2019) | | $61.76 \pm 0.08$ | $\mathbf{77.59 \pm 0.12}$ | $66.33 \pm 0.05$ | $81.44 \pm 0.09$ |
| MC (Park & Oliva, 2019) | | $61.22 \pm 0.10$ | $75.92 \pm 0.17$ | $66.20 \pm 0.10$ | $82.21 \pm 0.08$ |
| MC$^\dagger$ (Park & Oliva, 2019) | | $61.85 \pm 0.10$ | $77.02 \pm 0.11$ | $67.21 \pm 0.10$ | $82.61 \pm 0.08$ |
| MetaSGD + MetaNNF (ours) | center | $59.42 \pm 1.32$ | $70.24 \pm 0.73$ | $60.36 \pm 1.29$ | $75.08 \pm 0.66$ |
| MC + MetaNNF (ours) | | $\mathbf{63.40 \pm 1.30}$ | $76.12 \pm 0.68$ | $\mathbf{72.38 \pm 1.26}$ | $\mathbf{86.47 \pm 0.56}$ |
| LEO$^\dagger$ (Rusu et al., 2019) | multiview | $63.97 \pm 0.20$ | $79.49 \pm 0.70$ | — | — |
| MC$^\dagger$ (Park & Oliva, 2019) | | $64.40 \pm 0.10$ | $80.21 \pm 0.11$ | — | — |
| MC + MetaNNF (ours) | multiview | $\mathbf{66.54 \pm 1.29}$ | $\mathbf{86.52 \pm 0.54}$ | — | — |

The experimental setups follow from the standard $M$-class $N$-shot few-shot classification protocol (Ravi & Larochelle, 2017; Finn et al., 2017). In particular, $\mathcal{D}_t^{\mathrm{trn}}$ per task $t$ consists of $M$ randomly drawn classes, each containing $N$ labeled data. The default task-specific model is a standard 4-block convolutional NN (CNN) (Vinyals et al., 2016). Each block of the CNN comprises a $3 \times 3$ convolution layer, a batch normalization layer, a ReLU activation, and a $2 \times 2$ max pooling layer. After the convolutional blocks, a linear regressor with softmax activation is appended to perform classification. Following the practices of (Park & Oliva, 2019; Flennerhag et al., 2020), the number of convolutional channels is set to 128 to improve its fitting capacity. Additionally, to be consistent with Theorem 3.6, Sylvester NFs are adopted in all the tests.

To illustrate the benefit of learning more expressive priors, the first test compares MetaNNF with other meta-learning algorithms having different prespecified priors using the 5-class miniImageNet dataset (Vinyals et al., 2016). As a plug-in prior model, our MetaNNF can be readily integrated with other meta-learning methods that adopt different task-level and meta-level optimizers. In this test, we implement MetaNNF with MAML (Finn et al., 2017) and MetaSGD (Li et al., 2017). The results are listed in Table 1, where the performance metric is the average classification accuracy on new tasks. It is seen that our MetaNNF outperforms all the competitors in terms of classification accuracy. This empirically confirms the superiority of data-driven priors over the prespecified pdfs, as well as the effectiveness of MetaNNF in learning an expressive prior. Moreover, a remarkable performance gain can be observed on the 1-shot dataset. This justifies the claim that prior can be particularly informative when the training data are extremely scarce. For an apples-to-apples comparison, methods that use pre-trained feature extractors or more complicated models (e.g., residual networks) are not included in this table. The compatibility of MetaNNF to these models will be demonstrated in the subsequent tests.

The second test evaluates MetaNNF on miniImageNet and tieredImageNet feature embeddings extracted using a pre-trained wide ResNet(WRN)-28-10 backbone (Rusu et al., 2019). Compared to

Table 3: Performance comparison of MetaNNF against meta-learning and metric-learning methods on the CUB-20-2011 dataset. For fairness, the backbone model is a 4-block CNN.

| Method | Type | 5-class CUB-200-2011 | |
| --- | --- | --- | --- |
| | | 1-shot (%) | 5-shot (%) |
| MatchingNet (Vinyals et al., 2016) | metric-learning | $45.30 \pm 1.03$ | $59.50 \pm 1.01$ |
| MAML (Finn et al., 2017) | meta-learning | $58.13 \pm 0.36$ | $71.51 \pm 0.30$ |
| ProtoNet (Snell et al., 2017) | metric-learning | $37.36 \pm 1.00$ | $45.28 \pm 1.03$ |
| RelationNet (Sung et al., 2018) | metric-learning | $58.99 \pm 0.52$ | $71.20 \pm 0.40$ |
| DN4 (Li et al., 2019) | metric-learning | $53.15 \pm 0.84$ | $\mathbf{81.90 \pm 0.60}$ |
| MattML (Zhu et al., 2021) | meta-learning | $66.29 \pm 0.56$ | $80.34 \pm 0.30$ |
| MAML + MetaNNF (ours) | meta-learning | $\mathbf{69.24 \pm 1.36}$ | $80.41 \pm 0.60$ |
| MetaSGD + MetaNNF (ours) | | $\mathbf{69.94 \pm 1.34}$ | $80.54 \pm 0.59$ |

the 4-block CNN, this model has a greater number of parameters and thus enhanced expressiveness. The results are summarized in Table 2, where MetaNNF is implemented with MetaSGD (Li et al., 2017) and MetaCurvature (MC) (Park & Oliva, 2019). In all test setups, MetaNNF brings about notable performance improvement compared to the corresponding baselines. This validates MetaNNF's effectiveness and flexibility as a plug-in prior module.

The last test assesses the performance of MetaNF on the CUB-200-2011 dataset (Wah et al., 2011). In contrast to the previous two datasets that contain nature images of distinct objects, this dataset specifically focuses on birds of various species. While the classification of nature objects primarily relies on low-level features such as shapes and colors, classifying various birds requires further recognition of high-level features including textures and segmentations. To learn these complicated features, the model needs to be either trained with sufficient data, or equipped with a powerful prior. Table 3 showcases the performances of different meta- and metric-learning methods on such a dataset. Again, our MetaNNF method is markedly effective on the 1-shot dataset where data are exceptionally limited. This highlights the significance of an expressive prior. For the 5-shot dataset where data are relatively abundant, its performance is also comparable to the state-of-the-art ones.

### 4.3 ABLATION STUDY

Next, ablation tests are conducted to analyze the performance gain of MetaNNF. The test is carried out on the miniImageNet dataset, with results gathered in Table 4. The first ablation investigates the impact of the advocated non-injective NFs over the injective ones. To ensure the injectivity of the Sylvester NF $f$, we follow the QR parameteriza-

Table 4: Ablation tests for MetaNNF.

| Ablation setup | 5-class miniImageNet | |
| --- | --- | --- |
| | 1-shot (%) | 5-shot (%) |
| - (baseline) | $\mathbf{59.10 \pm 1.52}$ | $\mathbf{71.48 \pm 0.68}$ |
| Injective NF | $56.72 \pm 1.46$ | $69.41 \pm 0.68$ |
| ReLU $\sigma$ | $56.54 \pm 1.46$ | $69.84 \pm 0.68$ |

tion recommended in (van den Berg et al., 2018). One can see the improved performance of non-injective NF due to its enhanced expressiveness, which numerically verifies Theorem 3.1 and Remark 3.4. The second ablation examines the influence of nonlinear function $\sigma$ in the Sylvester NFs. By changing the $\sigma$ from sigmoid to the popular ReLU activation, a degradation of empirical performance can be observed. This observation corroborates with Remark 3.10.

## 5 CONCLUSIONS AND OUTLOOK

An informative prior plays a crucial role in training a large-scale model with limited small-scale data. This work introduced a novel NNF model for learning an expressive task-invariant prior. By transforming a known pdf of a continuous random vector, the NNF model enables a large family of target pdfs. As a flexible plug-in prior model, our MetaNNF method offers enhanced prior expressiveness compared to existing meta-learning methods that rely on preselected prior pdfs. Numerical studies validate our theoretical analysis, and highlight the superior performance of the proposed method, especially when datasets are scarce. Our future research agenda includes i) investigation of more generic universal approximation theorems; ii) bilevel convergence analysis for the MetaNNF method; and, iii) implementation of MetaNNF with alternative NFs, backbone models, and meta-learning methods.

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
