# OpenReview forum: "Meta-Learning Universal Priors Using Non-Injective Normalizing Flows"
_ICLR.cc/2024/Conference — ICLR 2024 Conference Withdrawn Submission_

### Official Review · Reviewer_EPFW · 2023-10-30

**Soundness:** 3 good
**Presentation:** 3 good
**Contribution:** 3 good
**Rating:** 6
**Confidence:** 4

**Summary:**

This paper proposes meta-learning prior distribution of parameters where the task solver of the inner loop is assumed to be the maximum a posteriori (MAP) estimator. The gradient-based meta-learning such as MAML can also be interpreted as meta-learning mean vector of implicit Gaussian prior distribution. However, this paper argue the assumption of pre-defined form for prior distribution (e.g., Gaussian) can have limited expressiveness. To overcome this, this paper proposes non-injective normalizing flows for modeling the prior distribution. The experimental results show that proposed prior distribution consistently shows better performance than existing meta-learning methods.

**Strengths:**

- This paper is well-written and easy-to-follow. The algorithm table summarizes the overall logics to understand this paper.

- The motivation of modeling the expressive prior distribution for meta-learning is very intuitive.

- The experimental results show the efficacy of the proposed method over well-known benchmark datasets for meta-learning.

**Weaknesses:**

- I am curious about the novelty of this paper in relation to Sylvester NF. This paper proposes the use of non-injective normalizing flows, specifically Sylvester NF (van den Berg et al., 2018). It would be beneficial to discuss the differences between the proposed method and the approach presented by van den Berg et al. in 2018.

- This paper introduces Sylvester NF for $f: z \in \mathbb{R}^d \mapsto z' \in \mathbb{R}^d$, where $d$ is the dimensionality of neural network parameters ($\phi_t \in \mathbb{R}^d$). This limitation could hinder the scalability of the proposed method when dealing with an increasing number of parameters in neural networks.

**Questions:**

- The theoretical analysis in this paper primarily centers around the validity of non-injective normalizing flows, specifically Sylvester NF. Can you provide any theoretical analysis to help readers understand the necessity of the expressiveness of the prior distribution?

- Can you visualize the prior distribution that has been meta-learned with image data? I am curious to know whether the meta-learned prior distribution exhibits multi-modality or not.

---

### Official Review · Reviewer_8K48 · 2023-11-01

**Soundness:** 2 fair
**Presentation:** 2 fair
**Contribution:** 2 fair
**Rating:** 5
**Confidence:** 3

**Summary:**

The work proposes to learn an universal prior powered by non-injective normalizing flows for effective meta learning.

**Strengths:**

The authors offered relatively detailed introduction on the mathematical background, and the derivation foundation looks pretty solid. The experiment result looks better than state of the art.

**Weaknesses:**

The overall workflow of the description on theorem formulation is a bit scattered. Work is needed to make the description on theorems more logical with each other. Also, please check if some of the math concepts are really necessary in the formulation.

In section 3.1, clarification is needed on why Pz'(z') defined based on the bijection function f is a good approximation of q.

Is some of the notions described in the article closely related to prior design of meta learning? e.g. what is the strength of pushforward measure? why are we choosing Borel  sigma-algebra for \mathcal{B} in theorem 3.1 and not other algebras?

Given the significant improvement of performance on 1-shot few shot learning of miniimagenet, I would assume significantly richer information is encoded in this prior. How does this proposed prior hold this much additional information?

In section 3.3, why “optimization of the transformation parameter \theta_f” can be replaced with "optimize the latent vector z_t", given \theta_f and z_t are totally different in meaning.

minors:

What is “meaning fitting ability”? Explanation needed.

"when data are scarce or costly to annotate. Examples of such applications include machine translation", the translation task don't have this data scarcity problem.

"Conventional approaches to meta-learning have relied on hand-crafted techniques to extract prior
knowledge" some recent works to support this.

**Questions:**

In section 3.3, why “optimization of the transformation parameter \theta_f” can be replaced with "optimize the latent vector z_t", given \theta_f and z_t are totally different in meaning.

What is “meaning fitting ability”? Explanation needed.

---

### Official Review · Reviewer_VGG6 · 2023-11-04

**Soundness:** 2 fair
**Presentation:** 2 fair
**Contribution:** 1 poor
**Rating:** 3
**Confidence:** 3

**Summary:**

The paper proposes to use normalizing flow for meta-learning. They first argue that for generic probability measures, the "flow" that pushes a
reference measure to the other is not necessarily injective. For this non injective Sylvester NFs are introduced, which are basically Sylvester NFs without constraints. Then the normalizing flow is applied to meta learning: This is a two step algorithm where first the samples for each task are propagated by the flow and then the flow is updated via an average over all tasks. The effciency of the algo is evaluated on several meta learning benchmarks.

**Strengths:**

Disclaimer: I am very familiar with the NF literature but not at all knowledgeable in terms of meta learning. Please bear that in mind.

The paper is well-written. The experimental results seem strong, I also like the ablation studies. The proofs are easy to follow.

**Weaknesses:**

The paper unfortunately has some weaknesses, some of them might be to my misunderstandings tho.

1) the literature review in the normalizing flow direction is not complete. Some of the paper deals with the tails of pdfs. A detailed account of NFs trying to model heavy (light)-tailed distributions in [1]. The paper furthermore ignores important recent literature on NFs which is very related, for instance [2], [3].

2) I am not sure about the novelty of theorem 3.1. It is presented and described in a way that lets one think this might be novel, but I think this is an easy consequence of the monge map being the gradient of a convex function. The Monge map should exist under these conditions, and as the gradient of a convex function it should also be what the authors call weakly increasing. (Also I think the authors should define weakly increasing as this is not standard defintions. It is only explained in the proof)

Or to see it differently look at the Explanation of [1], where they argue how to construct the well-known Knothe-Rosenblatt transform, which should also give the same results as theorem 3.1.

3) I dont understand non-injective Sylvester Flows. In the usual Sylvester Flow paper a more specific structure is used to ensure invertibility (they find conditions on the diagonal of the triangular matrices so that it is invertible). Here this looks more like a ResNet. Also the algorithm does not use any NF related concepts such as maximum likelihood training and only forward evals are needed. I am not sure whether this should even be called a normalizing flow? the only similarity i see is using a prior/latent distribution.

4) I find it quite weird that the validation data is also used when optimizing the Sylvester NF. This however might be very common practice in meta learning, which I am not familiar with.

5) The motivation for non injective flows in general is not super convincing to me. There have been some results (for specific NFs) that they are universal distribution approximators, i.e., having two lebesgue densities one can at least find a sequence of NFs with distance between the distributions going to 0, see e.g. [4].

[1] Tails of Lipschitz Triangular Flows, Jaini et al

[2] Generalization of the Change of Variables Formula with Applications to Residual Flows, Koenen et al

[3] MetaFlow: A Meta Learning With Normalizing Flows Approach For Few Shots Learning, Ahmed et al

[4] Coupling-based invertible neural networks are universal diffeomorphism approximators, Teshima et al

**Questions:**

See weaknesses.